# Active layer depth and soil properties impact specific leaf area variation and ecosystem productivity in a boreal forest

Carolyn G. Anderson[ORCID][1][¤]*, Ben Bond-Lamberty[2], James C. Stegen[1]

1 Pacific Northwest National Laboratory, Richland, Washington, United States of America, 2 Pacific Northwest National Laboratory, Joint Global Change Research Institute, College Park, Maryland, United States of America

¤ Current address: Stockbridge School of Agriculture, University of Massachusetts Amherst, Amherst, Massachusetts, United States of America
* cganderson@umass.edu

**Data Availability Statement:** All information needed to access the data underlying this study is in the Supporting information.

## Abstract

Specific leaf area (SLA, leaf area per unit dry mass) is a key canopy structural characteristic, a measure of photosynthetic capacity, and an important input into many terrestrial process models. Although many studies have examined SLA variation, relatively few data exist from high latitude, climate-sensitive permafrost regions. We measured SLA and soil and topographic properties across a boreal forest permafrost transition, in which dominant tree species changed as permafrost deepened from 54 to >150 cm over 75 m hillslope transects in Caribou-Poker Creeks Research Watershed, Alaska. We characterized both linear and threshold relationships between topographic and edaphic variables and SLA and developed a conceptual model of these relationships. We found that the depth of the soil active layer above permafrost was significantly and positively correlated with SLA for both coniferous and deciduous boreal tree species. Intraspecific SLA variation was associated with a fivefold increase in net primary production, suggesting that changes in active layer depth due to permafrost thaw could strongly influence ecosystem productivity. While this is an exploratory study to begin understanding SLA variation in a non-contiguous permafrost system, our results indicate the need for more extensive evaluation across larger spatial domains. These empirical relationships and associated uncertainty can be incorporated into ecosystem models that use dynamic traits, improving our ability to predict ecosystem-level carbon cycling responses to ongoing climate change.

## Introduction

The boreal forest is changing rapidly with climate change [1]. Permafrost soil underlaying the boreal is currently degrading and in many places predicted to disappear entirely, by the end of the 21st century in Alaska [2] and other circumpolar regions [3]. Permafrost thaw affects ecosystem carbon, water, and nutrient cycling [4–6], which are expected to, in turn, produce shifts in tree cover and canopy physiology [7]. Moreover, permafrost thaw has been shown to be a threshold for these environmental shifts [8, 9].

**Funding:** The research was conducted under the Laboratory Directed Research and Development (LDRD) Program at Pacific Northwest National Laboratory, a multiprogram national laboratory operated by Battelle for the U.S. Department of Energy under Contract DE-AC05-76RL01830. A portion of the research was performed using EMSL, a DOE Office of Science User Facility sponsored by the Office of Biological and Environmental Research and located at the Pacific Northwest National Laboratory. The funders had no role in study design, data collection and analysis, decision to publish, or preparation of the manuscript.

**Competing interests:** The authors have declared that no competing interests exist.

Phenotypic plasticity allows plants to adapt to environmental shifts, resulting in intraspecific trait variation. A particularly variable trait is specific leaf area (SLA). SLA—leaf area per unit dry mass—is a trait that corresponds with differences in leaf structure associated with photosynthesis [10] and importantly ecosystem carbon gain [11]. SLA has been used in numerous meta-analyses to predict leaf physiology and other functional traits [12].

Understanding the consequences of permafrost thaw on SLA variation and ecosystem productivity is particularly important, as shifting environmental gradients may impact intraspecific trait variation, with potentially large consequences on carbon accumulation across the landscape. In environments with optimal soil resources (e.g., water, nutrients), for example, plants can produce more leaf biomass with high SLA, maximizing carbon gain per unit leaf mass [13, 14]. Conversely, in suboptimal resource environments, small and thick leaves (i.e., with low SLA) allow plants to maximize leaf longevity. The relatively thick (low SLA) leaves of black spruce, a typical boreal evergreen conifer, last an average of 50–60 months, compared to the 4–6 month life span of the relatively thinner (high SLA) leaves of boreal deciduous species [15].

In boreal systems underlain by permafrost, the thaw depth of the seasonally-thawed active layer [9] is coupled to soil moisture and nutrient availability, and is hypothesized to govern leaf area and plant productivity [16–18]. While SLA of boreal vegetation has been shown to vary with environmental conditions, including tree species and soil resources [19–22], the effect of permafrost on SLA variation is not well understood, due in large part to the lack of empirical SLA data across a broad range of permafrost conditions.

Global analyses reveal that specific leaf area varies with climatic and edaphic gradients [23, 24]. In contrast, SLA variation within a species is less well understood. Intraspecific SLA variation has been shown to contribute significantly to total trait variability [20, 25–27], and used to understand local and regional community assembly processes and explain the coexistence of multiple species across environmental gradients [28–30].

Because SLA is linked to forest productivity through photosynthetic potential, understanding the environmental controls on SLA variation is also important for predictive ecosystem modeling [31], especially in climate-sensitive permafrost systems. Most ecosystem models assign a specific leaf area value based on plant functional types [32], and efforts have been made to improve mean canopy estimates of SLA [33]. While this captures variation in specific leaf area as a function of climate or species, these fixed trait-based approaches miss the variation in a trait within a given plant functional type. Without data on intraspecific trait variation, it is not clear whether these models can be successful.

In this study we examined intraspecific SLA variation and associated topographic and edaphic factors across a permafrost and vegetation transition within an Alaskan boreal forest. We hypothesized that (i) SLA would be significantly correlated with active layer depth, which governs the availability of soil resources such as water and nutrients, and (ii) intraspecific SLA variation across the permafrost transition would in turn positively correlate with aboveground net primary production. This is an exploratory study to begin understanding SLA variation in a permafrost forest ecosystem. Our results indicate significant influences of environmental features across the permafrost transition zone, indicating the need for more extensive evaluation of SLA and forest production across larger spatial domains in forested ecosystems.

## Materials and methods

### Site description

The field component of this research took place in the Caribou-Poker Creeks Research Watershed (CPCRW), a 104 km$^2$ basin located in the Yukon-Tanana Uplands northeast of Fairbanks, AK, and part of the Bonanza Creek LTER (http://www.lter.uaf.edu/research/study-

sites-cpcrw). This fieldwork was conducted under verbal permission of Jamie Hollingsworth, Site Manager of the Bonanza Creek LTER. Mean annual precipitation is 400 mm, about one-third of which falls as snow; mean annual temperature is -2.5˚C [34]. The watershed's lowlands and north-facing slopes are dominated by black and white spruce (*Picea mariana* and *Picea glauca*, respectively), feathermoss (*Pleurozium schreberi* and others), and *Sphagnum spp.*; the drier south slopes tend to be deciduous with a mixture of quaking aspen (*Populus tremuloides*), paper birch (*Betula neoalaskana*), and patches of alder (*Alnus crispa*).

In 2014 we established six replicate 75 m east-facing transects along a vegetation and permafrost gradient. To efficiently characterize spatial variation across north-south and east-west dimensions, we used a cyclic sampling design, a scheme that uses a repeated pattern of sampling points across space to allow comparison between sampling pairs at multiple distances (S1 Fig in S1 File) [35]. The transects were centered on 65.1616˚ N 147.4859˚ W at 248–266 m asl, with the east end of each transect dominated by black spruce in continuous, valley-floor permafrost, and the west end upslope dominated by paper birch with no permafrost within 150 cm. Both spruce and alder co-occur across the entire sampling transect. The forest in this study site was at least 90 years old, based on tree rings taken at the stem base of several of the largest white spruce (*Picea glauca*). Measured across the transects and inclusive of spruce, birch, and alder, stand density varied greatly across the site (overall 4890 trees ha$^{-1}$ ± 3290 standard deviation), with relatively low density at the top and bottom of the hillslope and a densely vegetated transition zone in the middle area, where black and white spruce co-occur. Similarly, basal area also varied greatly (17.6 m$^2$ ha$^{-1}$ ± 15.5 standard deviation), generally increasing towards the upslope portion of the site. The soil is characterized as a silt loam in the Olnes (well-drained, top of hillslope) or Karshner (poorly-drained, bottom of hillslope) series [36].

## Specific leaf area analysis

In August 2015, mature leaf samples from the top one-third of the canopy were collected from alder, black spruce, and white spruce. Spruce and alder were selected for this study because they co-occurred across the entire sampling transect. Leaves were sampled at six locations (0, 15, 30, 45, 60, and 72 m, from E to W) along each transect, with up to 10 leaves per tree per species and location (S1 Fig in S1 File). To maintain field moisture, leaves were stored with wet tissues in a cooler or refrigerator until analysis. Projected leaf area was determined using a flatbed scanner (HP Digital Sender 9250c, 300 dpi) and ImageJ [37] version 1.48. Hue and saturation were set at 255, and brightness at 170 for broadleaf species and 180 for needleleaf species. The default thresholding method was used, with color set to red. Hemisurface leaf area for nonflat spruce (*Picea spp.*) needles was calculated using equations from the literature [38]. Leaf dry mass was determined after drying to a constant mass in a forced-air oven at 60 ˚C, and specific leaf area (SLA, cm$^2$ g$^{-1}$) calculated by dividing leaf surface area by dry mass [39]. Black and white spruce species hybridize across vegetation transition zones such as the one in this study, making it difficult to distinguish them in some cases. For this reason, we pooled spruce species into a single vegetation type ("spruce"); this approach is consistent with a modeling-relevant approach [40]. However, we recognize the importance of species-specific data, and thus provide the full (raw) data with putative species tags (see Supporting information for data availability).

## Site characteristics and soil properties

Landscape slope, active layer depth (ALD), and soil cores were sampled in a cyclic sampling design to allow for efficient spatial analyses (S1 Fig in S1 File) [35]. Landscape slope was measured using a clinometer over a 2 m length centered at each soil sampling location. ALD was

measured from the soil surface every 2.5 m along each transect in September 2014 using a 150 cm probe; September thaw depths were used to capture maximum thaw. If permafrost was not reached at three consecutive positions moving upslope, ALD was assumed to be > 150 cm.

Along the southernmost three transects, soil cores were taken in September 2014 at each sampling location using a forest floor sampler [41]. After coring, thermistors were installed to measure depth-resolved soil temperatures, and the depth of the moss layer was recorded. To enable field-model comparisons, soil cores were subsampled for physicochemical and biological measurements at depth increments (1.75 cm, 6 cm, and 12 cm) corresponding with those used in the Community Land Model (CLM4.5) [40]. Soil moisture was determined by measuring the difference between fresh and dry mass after drying samples in a forced-air oven to a constant mass (g water per g dry soil), and pH was measured using 1:1 $CaCl_2$ mixture. Total soil carbon and nitrogen were determined using an Elementar Vario EL Cube Elementar (Elementar Analysensysteme GmbH, Hanau, Germany); C:N ratios were calculated from these data [24]. All site and soil data were linearly interpolated to match SLA data locations.

## Aboveground tree net primary production

Net primary production (NPP) was determined using tree cores taken from a representative sample of trees (birch, alder, black spruce, and white spruce) every 10 m along each transect. At each sample point, we cored 3–5 trees per species; sample discs were taken from trees too small to core. Cores were embedded into larger boards for protection, sanded, and scanned at 800 dpi using a flatbed scanner (Epson Workforce 840, Epson America Inc., Long Beach, CA, USA). Bark thickness and wood annual increments were measured to the nearest 0.001 mm using CooRecorder 7.6 (http://www.cybis.se/forfun/dendro/). For each of the most recent five years, ring width was used to calculate diameter in each year, and biomass estimated from species- and region-specific allometric equations [42]. For each year, tree NPP was computed as the difference between successive biomass estimates, and these values averaged to produce 5-year mean NPP for each species at each transect sample point. Annual mortality is typically low in mature boreal forests [43] and we did not correct for it in the calculated five-year window. The mean NPP was scaled to the site level using a tree inventory performed on all trees with a 5 m radius of each grid sample point within each transect. Here, we present both total NPP (combination of above-named species) and black spruce-specific NPP. The latter was performed on tree cores from the lower portion (E end) of the transects, where we were confident of the identity of black spruce trees (versus white spruce).

## Statistical analyses

All statistical analyses were performed using R [44] version 3.2.4. We assessed SLA variation across two scales: within and among individuals in a tree species. To partition the variation in these two scales, we fitted a general linear model to the variance across leaf and tree scales [27], with leaf nested in tree, and with separate analyses for each species (S1 Appendix in S1 File).

Because we have high-resolution spatial data, we used linear regressions (with SLA averaged by tree individual for each species) to test the hypothesized relationships between SLA and topographic and edaphic properties (Fig 1). Specifically, we performed Theil-Sen regressions as robust estimators against outliers, using the R function *mblm* in the 'mblm' package, version 0.12.1. Spearman's rank correlations were used to determine the significance and strength of relationships between ALD and slope and between ALD and SLA, using R function *cor.test* in the R 'stats' package, version 3.2.4. Where we saw relationships in the linear regressions, we used T- and F-tests to examine shifts in means and variances, respectively. When bivariate regressions showed nonlinear relationships, we used the function *segmented* in the R

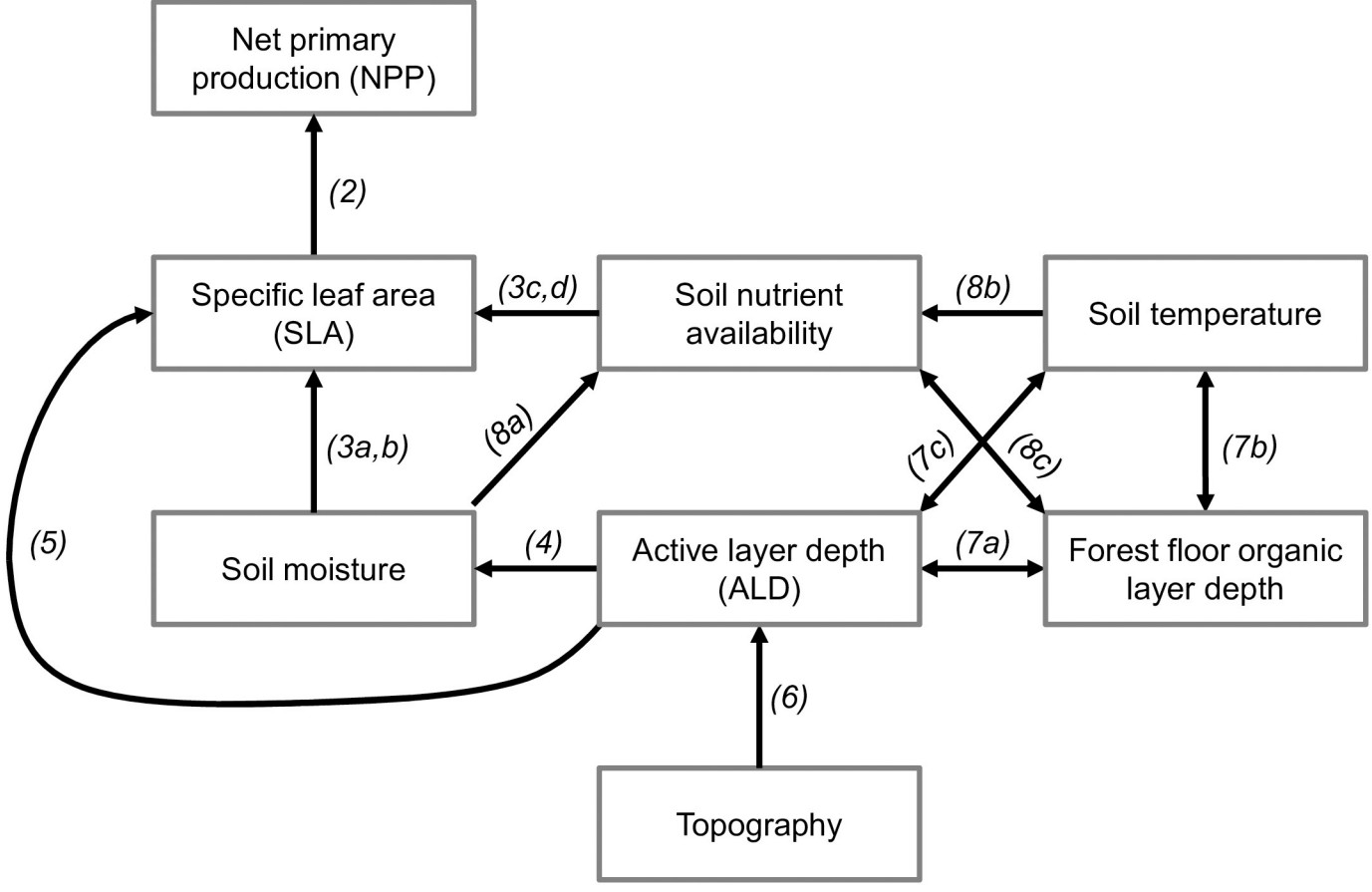

**Fig 1. Conceptual diagram linking topographic and edaphic features with specific leaf area (SLA) and net primary production (NPP)** [18]**.** Arrows indicate direct relationships; double-headed arrows indicate feedbacks. Arrow labels correspond to Figures.

'segmented' package to test for breakpoints and tested for significance of these breakpoints using the *davies.test* function.

## Results

### Inter- and intraspecific SLA variation

Specific leaf area (SLA) averaged by tree varied by over a factor of three between alder (n = 33 trees) and spruce (n = 36 trees), with high levels of intraspecific variation as well (Table 1). SLA for alder ranged from 121 to 364 $cm^2 g^{-1}$ (214 ± 66.1 $cm^2 g^{-1}$). Spruce SLA ranged from 37.1 to 88.7 $cm^2 g^{-1}$, (57.1 ± 9.61 $cm^2 g^{-1}$). Partitioning the variance of SLA into the leaf and

**Table 1. Minimum, maximum, and mean specific leaf area values ($cm^2 g^{-1}$) and coefficients of variation by species.**

| Species | n | Minimum | Maximum | Mean (s.d.) | Coefficient of Variation |
|---|---|---|---|---|---|
| Alder | 33 | 120.5 | 363.5 | 213.8 (66.1) | 0.309 |
| Spruce | 36 | 37.1 | 88.7 | 57.1 (9.6) | 0.168 |

Values in parentheses indicate 1 standard deviation.

tree scales revealed that most of the variation in SLA is between rather than within individual trees, for both alder and spruce (Table 2). Thus, all subsequent analyses were performed using tree-averaged SLA values, following the relationships in Fig 1.

## Impact of SLA on net primary production

Spruce-specific NPP was determined for the bottom four positions on the hillslope where we had corresponding data, and ranged from 23.1 to 364 g C m$^{-2}$ yr$^{-1}$. Because we did not determine alder-specific NPP across the whole spatial domain, SLA and NPP were compared only for spruce. Spruce-specific NPP was significantly higher as spruce SLA increased ($p < 0.001$, $R^2 = 0.42$, Fig 2a).

We hypothesized that SLA has a direct effect on NPP (Fig 1). However, it has been shown that soil nutrient status (e.g., soil C:N) may have a direct influence on NPP [45]. We found that spruce NPP and SLA were significantly correlated ($p < 0.001$, $R^2 = 0.42$, Fig 2a) as well as total NPP (from representative species, see Methods) and soil C:N ($p < 0.05$, $R^2 = 0.24$, Fig 2b).

## Relationship between soil resources and SLA

While our data show a negative linear relationship between both spruce and alder SLA and soil moisture the residual plots for the SLA and soil moisture linear regressions highlight potential nonlinearities in these relationships (S2 Fig in S1 File). Because of the small number of data

**Table 2. Variance partitioning of the full nested linear models on SLA across leaf and tree scales.**

| Species | Between-tree SLA variation (%) | Within-tree SLA variation (%) |
|---------|--------------------------------|-------------------------------|
| Alder | 93.7 | 6.3 |
| Spruce | 74.0 | 26.0 |

SLA data were natural log transformed prior to analysis. $n = 169$ leaves and 33 trees for alder; $n = 190$ leaves and 36 trees for spruce.

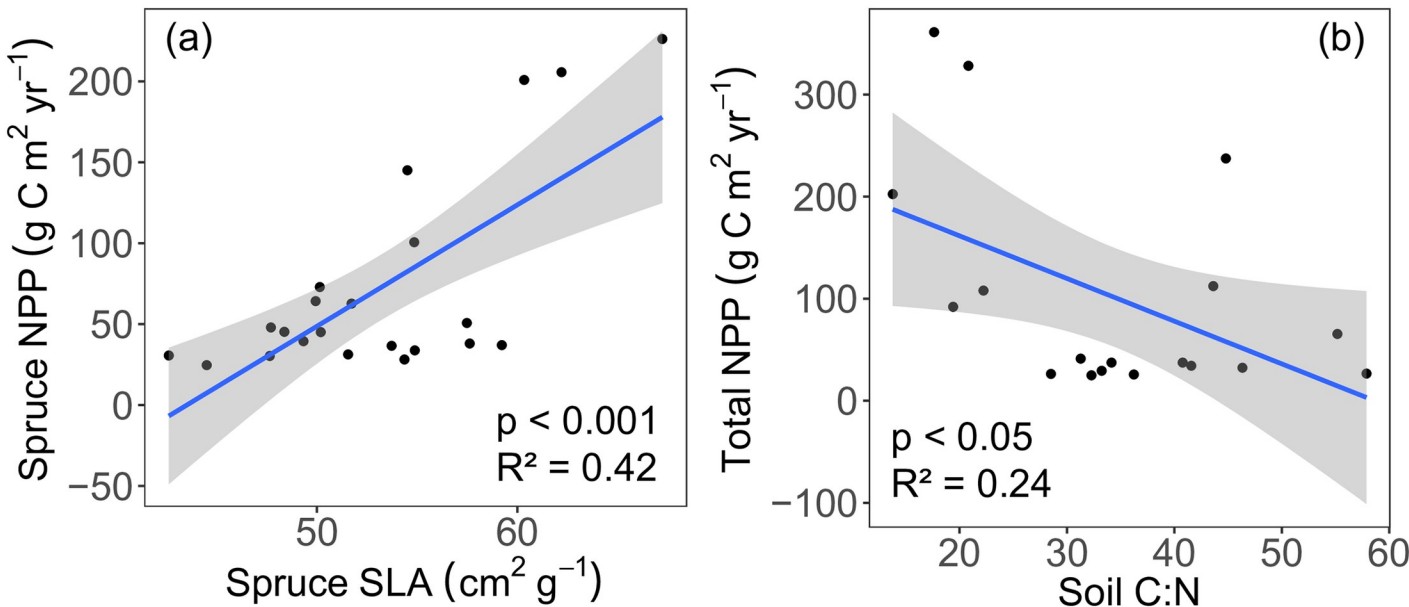

**Fig 2. Relationships between (a) spruce-specific SLA (cm$^2$ g$^{-1}$) and net primary production (NPP, g C m$^{-2}$ yr$^{-1}$), and (b) total NPP and soil C:N.** The blue lines represents the linear regression, and the shaded areas shows the 95% confidence level interval for predictions.

points in this study limits further interpretation of a nonlinear fit for these data, we therefore used linear regressions, which show higher SLA values correlated with lower soil moisture values ($p < 0.05$, $R^2 = 0.40$ for alder, Fig 3a; $p < 0.001$, $R^2 = 0.52$ for spruce, Fig 3b). For both alder and spruce, higher soil C:N values correlated with lower SLA values ($p < 0.05$; $R^2 = 0.30$ for alder, Fig 3c; $R^2 = 0.26$ for spruce, Fig 3d).

## Relationship between active layer depth and SLA

Across the entire field site, ALD ranged from 54 cm to above 150 cm (median 137.2 cm). Soil moisture at 6 cm ranged from 0.18–0.93 g g$^{-1}$ (mean = 0.68 g g$^{-1}$); we used soil data from 6 cm depth for comparisons with landscape and SLA data, as this depth is more relevant than surface soil to rhizosphere processes. Soil was drier at deeper ALD (for 6 cm, $p < 0.05$, Fig 4). For subsequent analyses, we used the ALD-moisture relationship to divide the data into two ALD classes: shallow ($< 140$ cm) and deep ($> 140$ cm) ALD; we used segmented regression to determine this breakpoint. Moisture values were marginally different between shallow and deep

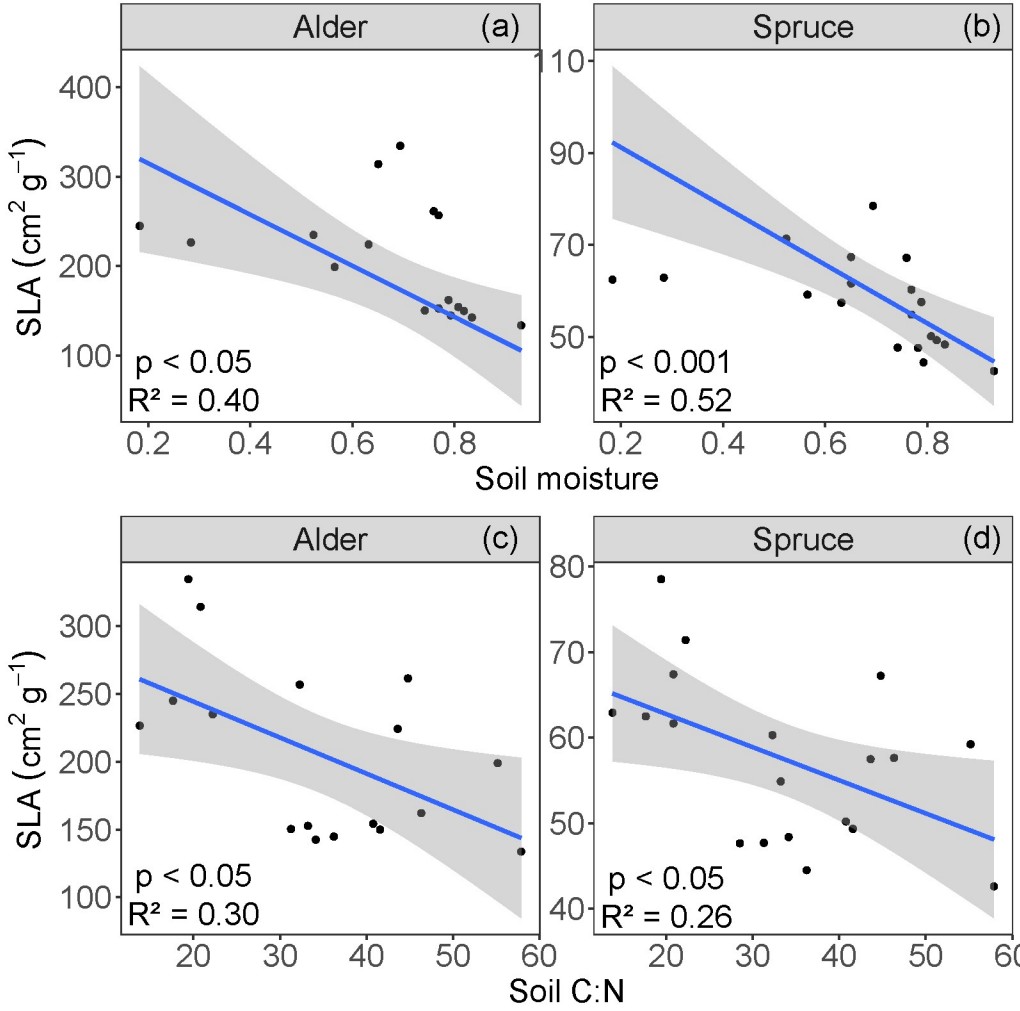

**Fig 3. Relationships between specific leaf area (SLA) and (a,b) gravimetric soil moisture and (b,c) C:N, for alder and spruce.** The blue line represents the regressions, and the shaded area shows the 95% confidence level intervals for predictions.

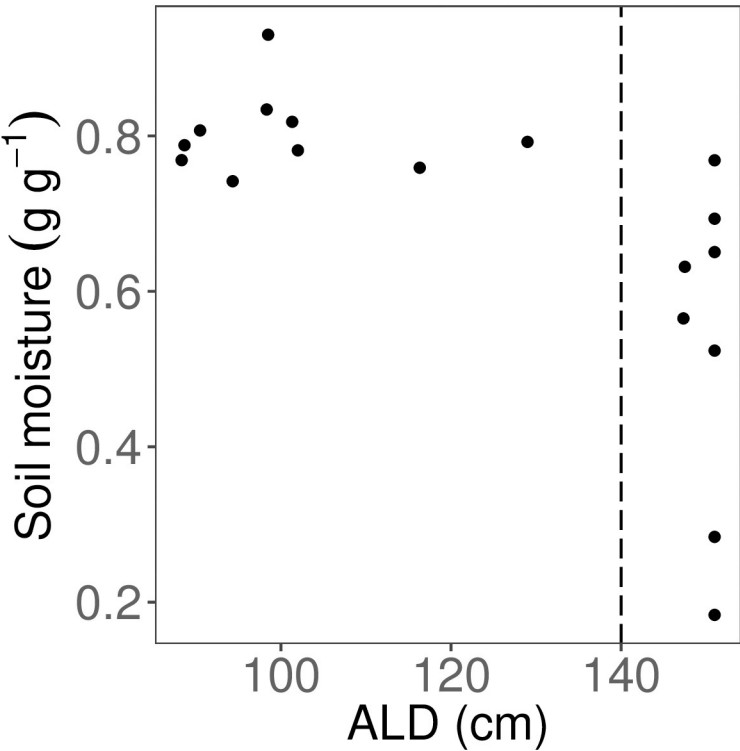

**Fig 4. Relationship between active layer depth (ALD) and gravimetric soil moisture.** Only 6 cm soil depth was used. Vertical dashed line indicates cut-off for shallow (< 140 cm) and deep (> 140 cm) ALD classes.

ALD classes (p = 0.0755). The variance in soil moisture differed significantly between the two ALD classes ($F_{9,7}$ = 0.0672, p < 0.001).

Mean alder SLA for deep ALD was 266 $cm^2 g^{-1}$, and for shallow ALD was 158 $cm^2 g^{-1}$. Mean spruce SLA for deep ALD was 62.0 $cm^2 g^{-1}$, and for shallow ALD was 52.1 $cm^2 g^{-1}$. SLA for both alder and spruce increased with thicker ALD (p < 0.001, $R^2$ = 0.62, ρ = 0.75 for alder, Fig 5a; p < 0.001, $R^2$ = 0.36, ρ = 0.62 for spruce, Fig 5b). For both alder and spruce, mean SLA corresponding with shallow and deep ALD were significantly different ($T_{30}$ = 8.94 for alder; $T_{34}$ = 5.01 for spruce; p < 0.001). There was no difference in SLA variances between the two ALD classes ($F_{15,16}$ = 0.557; p = 0.265 for alder; $F_{17,17}$ = 1.02; p = 0.971 for spruce).

## Topography influences on active layer depth

Across the field site, slope ranged from 13–56% (mean = 23.7%). ALD was positively correlated with landscape slope (p < 0.001, ρ = 0.63, Fig 6). Above 23% slope, ALD was consistently deeper than the maximum probe depth (150 cm); below this value, there was no significant correlation between ALD and slope (p = 0.157, rho = -0.31, Fig 6).

## Interactions between active layer depth, moss depth, and soil properties

Moss thickness varied from 3.0 to 23.5 cm (mean = 14.4 cm), and the 6 cm soil temperature at the time of coring ranged from 2.8 to 10.1°C (mean = 5.4°C). The moss layer was thickest in areas of shallow ALD (p < 0.05; $R^2$ = 0.27, Fig 7a). Soil temperature decreased with increased moss thickness (p < 0.05, $R^2$ = 0.41, Fig 7b), and increased with deeper ALD (p < 0.001, $R^2$ =

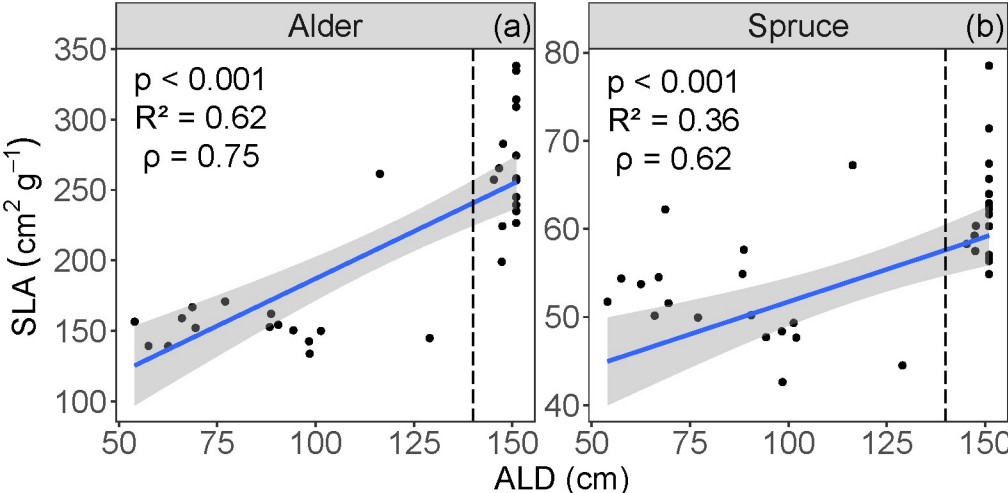

**Fig 5. Relationships between specific leaf area (SLA) and active layer depth (ALD), for (a) alder and (b) spruce.**
Vertical dashed lines indicate cut-off for shallow (< 140 cm) and deep (> 140 cm) ALD classes. The blue line represents the linear regression, and the shaded area shows the 95% confidence level interval for predictions.

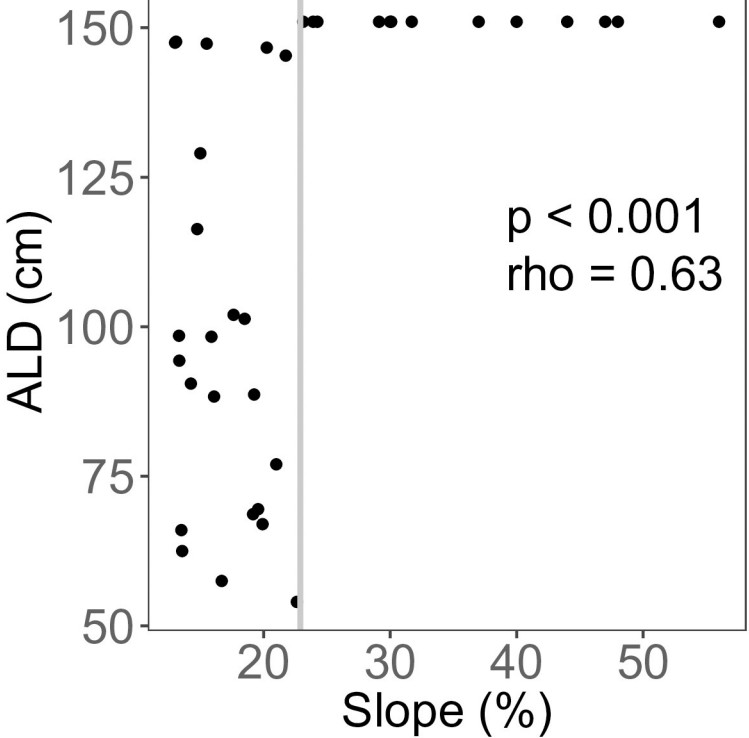

**Fig 6. Active layer depth (ALD) as a function of slope.** The whole data set was tested using Spearman's rank test ($p < 0.001$, $\rho = 0.628$). Below 23% slope (indicated by solid gray line), there was no significant correlation between ALD and slope ($p = 0.32$, $\rho = -0.305$).

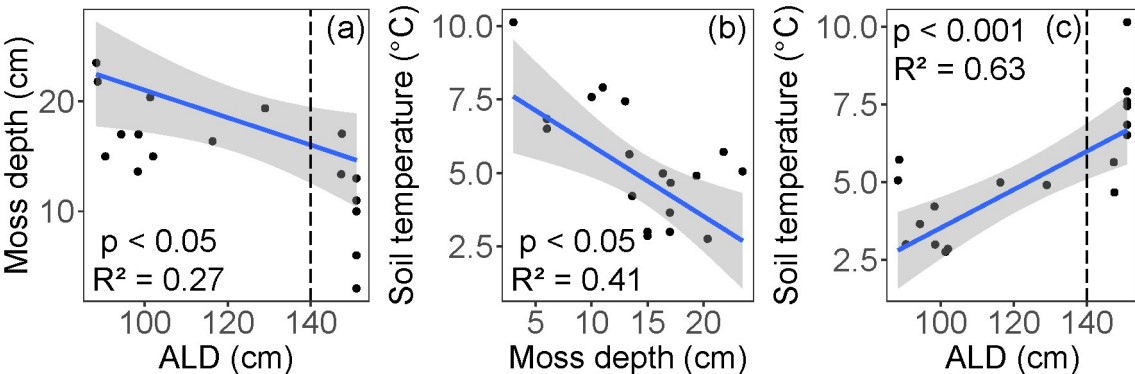

**Fig 7. Interactions of (a) the moss layer and active layer depth (ALD), (b) soil temperature and moss depth, and (c) soil temperature and ALD.** Vertical dashed lines in panels (a) and (c) indicate cut-off for shallow ($< 140$ cm) and deep ($> 140$ cm) ALD classes. The blue line represents the linear regression, and the shaded area shows the 95% confidence level interval for predictions.

0.63, Fig 7c). The means of both moss depth ($T_{12}$ = -4.11, p < 0.05) and soil temperature ($T_{12}$ = 4.56, p < 0.001) in the shallow and deep ALD classes were significantly different.

Soil C:N at 6 cm depth ranged from 13.8 to 57.9 (mean = 34.4). Soil C:N values were significantly higher in wetter soil conditions associated with shallower ALD (p < 0.001, $R^2$ = 0.40, Fig 8a). Soil C:N at 6 cm decreased with increased soil temperatures (p < 0.05, $R^2$ = 0.16, Fig 8b), and increased with thicker moss layers (p < 0.05, $R^2$ = 0.46, Fig 8c).

## Discussion

### Ecosystem productivity implications of intraspecific SLA variation

The empirical data presented here suggest that in permafrost-affected systems, the depth of the active layer can function as a threshold for various soil parameters, influencing plant traits such as SLA (Fig 5). SLA is directly associated with leaf-level photosynthesis, and has been shown to have a direct positive correlation with photosynthesis and productivity [14]. Here, we show that across a permafrost transition, a doubling in black spruce SLA corresponds to a much larger (five-fold) increase in NPP (Fig 2a). Further, our data suggest a direct connection between SLA and NPP (Fig 2a) and an indirect and weaker influence of soil C:N on NPP (Fig

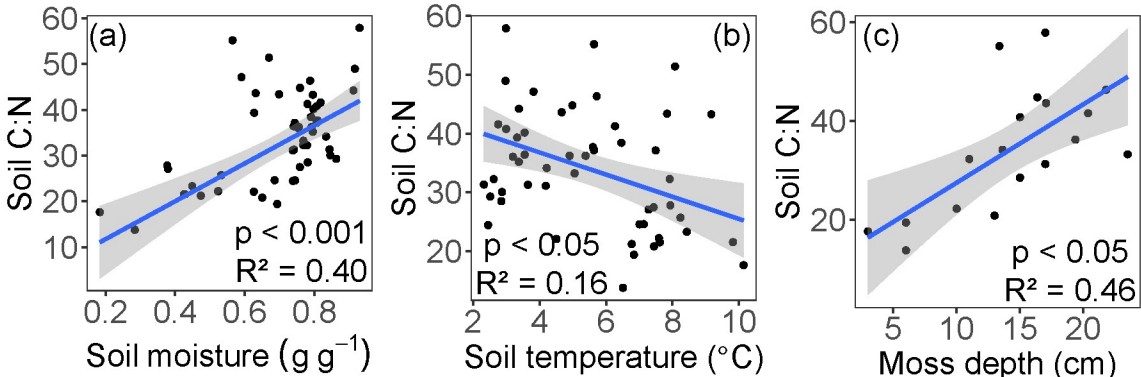

**Fig 8. Relationships between soil C:N and (a) soil moisture, (b) soil temperature, and (c) moss depth.** All soil depths were used in panels (a) and (b) (1.75 cm, 6 cm, 12 cm). Only 6 cm soil depth was used in panel (c). The blue line represents the linear regression, and the shaded area shows the 95% confidence level interval for predictions.

2b). Because the control on maximum SLA is mediated by ALD (Fig 5a and 5b), when permafrost thaws, the nonlinear relationship between specific leaf area and NPP represents an avenue for the ecosystem to gain carbon. This result is especially relevant to boreal systems, where climate change is expected to appreciably thaw the permafrost [2, 9]. This suggests that as the active soil layer deepens, carbon cycling will undergo nonlinear acceleration, a phenomenon called a tipping point.

In addition, the deepening of the active layer may accelerate belowground decomposition and thereby release carbon to the atmosphere [3]. Under warming conditions, while the released carbon might in principle be more readily fixed through the parallel increase in photosynthesis, analyses in warming boreal and arctic systems suggest that, in fact, respiration typically increases faster than photosynthesis [46]; this is also consistent with a recent global synthesis [47]. Ecosystem-scale carbon cycling responses will however depend on specific ecosystem parameters such as water balance, soil carbon composition, and carbon bioavailability [48, 49]. The balance between respiration and productivity warrants further exploration in permafrost-affected ecosystems sensitive to climate change.

## Influence of active layer depth on SLA variation

We found that while soil moisture does have a negative linear relationship with SLA (Fig 3a and 3b), this relationship has potential nonlinearities (S2 Fig in S1 File), although we note that the small number of data points at low moisture conditions limits this interpretation. SLA and moisture have been shown to be positively correlated in systems with low soil moisture values $(0.05–0.30 \text{ g g}^{-1})$ [29]. In contrast, studies in boreal systems with more consistently saturated conditions $(0.60+ \text{ g g}^{-1})$ have shown that SLA is constrained at high-moisture sites with shallow ALD [19, 22], likely due to saturated conditions in which the anaerobic environment leads to reduced root conductance [50], limited nutrient availability, and generally poorer conditions for plant growth. The relatively large range of soil moisture included our study site $(0.18–0.93 \text{ g g}^{-1})$ may therefore explain the nonlinearities in the relationship between SLA and soil. We reiterate that SLA data in boreal systems more evenly distributed across a wide range of soil moisture values are needed to unambiguously distinguish between a negative linear relationship (more soil moisture always means lower SLA, as the negative correlations in our data show) versus a quadratic one (implying a moisture optimum).

Significant variation in SLA was also explained by soil C:N (Fig 3c and 3d). Global interspecific plant trait studies have shown that higher soil C:N correlates with lower SLA [24]. In our study looking at intraspecific SLA variation across an ALD gradient, we find that higher soil C:N also correlates with lower SLA. This is an important plant-soil relationship in the context of predictive trait modeling, especially in N-limited high latitude systems subject to permafrost thaw and thus changes in water, carbon, and nutrient cycling [4–6]. If permafrost thaw increases the pool of available soil N (i.e., lower soil C:N), then photosynthetic nitrogen demands could be alleviated, potentially increasing SLA and site productivity; furthermore, if this response to changes in soil C:N varies within a species, current fixed-trait models will not capture these changes.

For both species, we found that ALD is highly correlated with SLA, with an approximate two-fold increase in species-specific SLA values from shallow to deep ALD locations (Fig 5). Further, ALD has a thresholding effect on intraspecific SLA, whereby a relatively small change in ALD corresponds to a large change in SLA (Fig 5). While SLA at the landscape scale is controlled by soil resources (moisture and nutrients) [51], in this boreal system this control is mediated by ALD, which constrains SLA (Fig 5). Our data extend the findings of SLA dynamics in permafrost systems [19, 22, 52] by providing a broad range of continuous ALD across a

small-scale and critical soil transition zone, while also examining a network of soil and landscape influences to understand controls on SLA (Fig 9).

Given the structure of our hypothesized relationships between SLA and environmental correlates (Fig 1), we considered path analysis (a type of structural equation modeling, SEM). However, we have insufficient data given the number of variables that would be contained within an SEM. For example, separating the soils dataset by tree species, we have 18 data points, and the number of parameters is 7 (excluding NPP, Fig 1); our number of samples per parameter is 2.6, which is on the low end of sample adequacy [53]. We instead performed bivariate linear regressions as detailed above, using our hypothesized relationships (Fig 1) to construct a conceptual diagram (Fig 9). Further studies should consider increasing sample size to allow for SEM analysis. Because we evaluated relationships based on *a priori* hypothesized relationships (Fig 1), we did not perform a global selection model to explain variation in SLA. As a caveat, our data describe one spatial domain (75m x 75m), and further study is needed to confirm if the relationships between SLA and active layer depth and soil properties apply at other spatial scales.

We assume here that the high degree of SLA variation among individuals in a given species (Table 1) is due to phenotypic plasticity in response to environmental gradients, rather than genetic variation. Given the relatively small spatial domain of our study site (ca. 75 m x 75 m) and the steep topographic, vegetation, and permafrost gradients encountered, we believe this is a reasonable assumption. While we do not attempt to resolve all ecological scales that influence total trait variation (i.e. leaf, canopy-level, tree, species, plot, and site) [27], our results suggest that it is at the stand or plot level that environmental filters (e.g., ALD, soil C:N) act on SLA. This is similar to studies in tropical [27] and Mediterranean [29] systems, and provides guidance in the level at which intraspecific trait variation is ecologically relevant.

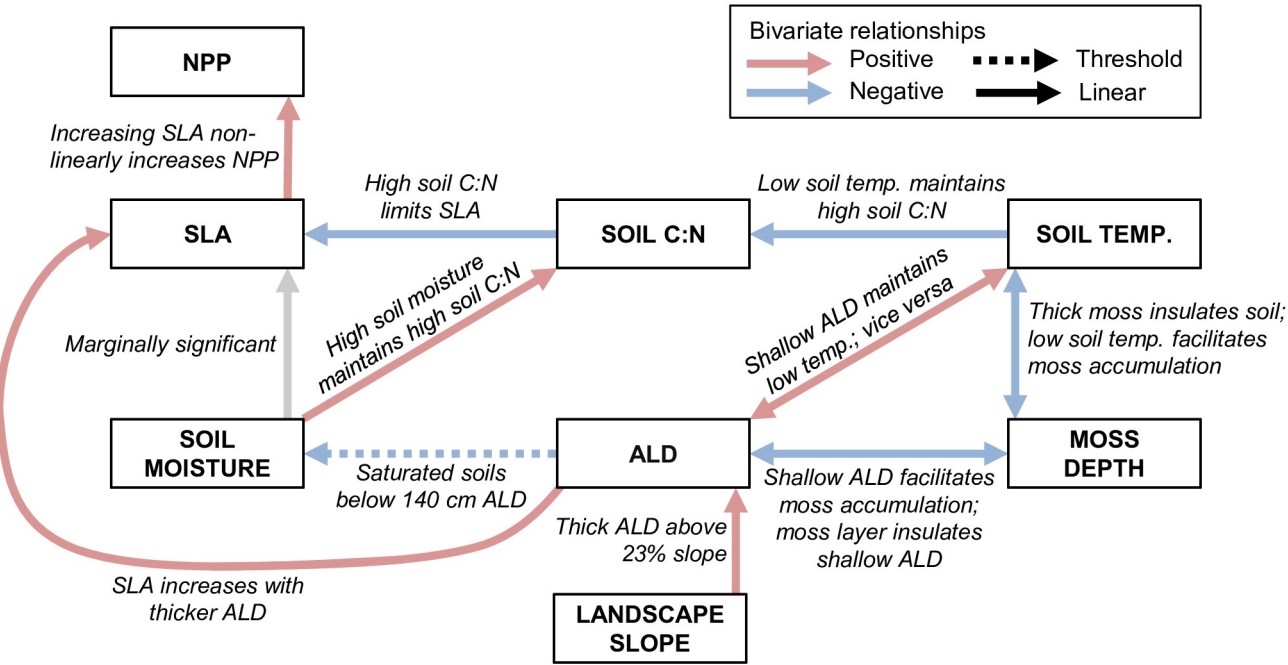

**Fig 9. Revised conceptual diagram (not formal SEM) based on this study's results linking topographic and edaphic features, including active layer depth (ALD), with specific leaf area (SLA) and net primary production (NPP) [18].** Relationships are based on bivariate regressions. Solid lines indicate statistically significant linear relationships and dashed lines indicate a nonlinear threshold relationship. Arrow color indicates direction of correlation: red is positive; blue is negative; gray represents a marginally significant linear relationship.

## Relationships between active layer depth and edaphic and topographic features

The relationships between permafrost and topographic and edaphic features are well documented [54, 55]. Landscape slope and aspect affect the amount of incoming solar radiation, which in turn influences soil thermal conditions and depth to permafrost [18, 48, 56]. We controlled for aspect in this study (all transects are east-facing), allowing us to isolate the effect of slope on ALD. We found a threshold in the slope-ALD relationship, with permafrost not encountered at 150 cm depth at slopes greater than 23% (Fig 6). This is not unexpected, as deeper ALD is predicted on steeper slopes due both to increased drainage and to high solar radiation inputs in high latitude systems [18].

Because permafrost is a physical barrier to water drainage, the shallower active layers generally maintain high soil moisture conditions [18]. We found that shallow active layers (< 140 cm) constrain soil moisture values to near saturation and lowered their variance (Fig 4). Together, high moisture and low temperatures of permafrost-associated soils limit decomposition, thus maintaining high C:N ratios in the thick moss layer and limiting nutrient availability [5] (Fig 8). Past the observed ALD threshold of 140 cm, soil moisture and thermal conditions shift rapidly (Figs 4 and 7), likely because deeper ALD (often found at steeper slopes) promotes both water drainage and heat advection by pore water flow [57, 58]. Together, these conditions likely favor aerobic decomposition of soil organic matter and increase available nutrients for plant uptake.

In boreal systems, both the surface moss and organic soil layer play important roles in soil thermal dynamics, nutrient cycling, and ecosystem carbon accumulation [59]. Specifically, the moss and organic layers act as insulation to maintain low soil temperatures and shallow ALD [9, 18, 54, 60], an effect modulated by soil moisture [61]. In support of this thermal regulation mechanism, our data show expected relationships between moss depth, temperature, and ALD (Fig 7). In turn, low soil temperatures maintained by the moss layer can limit decomposition and maintain high soil C:N (Fig 8), and thus limit the availability of soil nutrients [18].

## Implications for earth system models

Most ecosystem process models represent vegetation by plant functional types (PFTs), each of which has fixed photosynthetic parameters such as SLA that collectively control PFT photosynthesis, allocation, mortality, and decomposability. In the Community Land Model (CLM4.5), for example, the assigned SLA value for each PFT is mechanistically linked to photosynthesis through leaf N concentration [40, 62]. Modifications to CLM have been proposed that improve representations of leaf traits, nitrogen availability, and plant productivity, with large implications for gross primary production [63]. However, averaging plant trait parameters is problematic, because for several important traits there is more variation within PFTs than between PFTs [12, 64, 65].

Extensive plant trait databases allow for the study of intraspecific variation across species and PFTs [65], which has led to calls for inclusion of such variation in ecological studies [66]. In turn, ecosystem models have been developed which incorporate trait plasticity into the existing PFT representation based on soil and climate relationships [67–69], and also approaches that utilize dynamic functional trade-offs rather than PFTs [70]. Model simulations with dynamic traits based on empirical trait-environment relationships show that high latitude systems become a stronger carbon sink through the next century, due in part to increased productivity [67]. Knowledge of empirical patterns like those shown in this study—whereby high intraspecific SLA variation leads to nonlinear increases in productivity (Fig 2a)—is necessary to increase the predictive power of these models and provide benchmark data

with which to test their performance. Such knowledge is especially crucial in climate-sensitive high-latitude systems, where permafrost thaw impacts drivers of plant trait variation [71].

## Conclusions

The empirically-derived relationships presented in this exploratory study (Fig 9) can guide the structural form and associated uncertainty of environmental controls on and consequences of SLA variation in process models. Our results indicate the need for more extensive evaluation of SLA variation and its impacts on forest ecosystem production, particularly across larger spatial domains that are replicated across multiple hillslopes distributed through non-contiguous permafrost ecosystems. Together, our study connects fundamental understanding of the linkages between SLA and system features with a meaningful way to incorporate them into the latest Earth System Models and a predictive framework for forest management under climate change.

## Supporting information

**S1 File.**
(DOCX)

## Acknowledgments

We are grateful to Jamie Hollingsworth for technical support at the Caribou-Poker Creeks Research Watershed (CPCRW), part of the Bonanza Creek Long-Term Ecological Research site. The research was conducted under the Laboratory Directed Research and Development (LDRD) Program at Pacific Northwest National Laboratory, a multiprogram national laboratory operated by Battelle for the U.S. Department of Energy under Contract DE-AC05-76RL01830. A portion of the research was performed using EMSL, a DOE Office of Science User Facility sponsored by the Office of Biological and Environmental Research and located at the Pacific Northwest National Laboratory. The authors declare no conflict of interest.

## Author Contributions

**Conceptualization:** Carolyn G. Anderson, Ben Bond-Lamberty, James C. Stegen.

**Formal analysis:** Carolyn G. Anderson.

**Funding acquisition:** James C. Stegen.

**Methodology:** Carolyn G. Anderson, Ben Bond-Lamberty, James C. Stegen.

**Visualization:** Carolyn G. Anderson.

**Writing – original draft:** Carolyn G. Anderson.

**Writing – review & editing:** Carolyn G. Anderson, Ben Bond-Lamberty, James C. Stegen.

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
