## [Decision Letter · Decision Letter 0]

23 Jun 2020

PONE-D-20-10890

Controls on and consequences of specific leaf area variation with permafrost depth in a boreal forest

PLOS ONE

Dear Dr. Anderson,

Thank you for submitting your manuscript to PLOS ONE. After careful consideration, we feel that it has merit but does not fully meet PLOS ONE’s publication criteria as it currently stands. Therefore, we invite you to submit a revised version of the manuscript that addresses the points raised during the review process.

We look forward to receiving your revised manuscript.

Kind regards,

Dafeng Hui, Ph.D.

Academic Editor

PLOS ONE

Additional Editor Comments:

I now have two reports from expert reviewers. Both reviewers considered this study interesting, but also raised some concerns with current manuscript. A revision is needed before it can be recommended for publication.

Journal Requirements:

Reviewers' comments:

Reviewer's Responses to Questions

**Comments to the Author**

1. Is the manuscript technically sound, and do the data support the conclusions?

Reviewer #1: Yes

Reviewer #2: Yes

2. Has the statistical analysis been performed appropriately and rigorously? 

Reviewer #1: Yes

Reviewer #2: No

3. Have the authors made all data underlying the findings in their manuscript fully available?

Reviewer #1: Yes

Reviewer #2: Yes

4. Is the manuscript presented in an intelligible fashion and written in standard English?

Reviewer #1: Yes

Reviewer #2: No

5. Review Comments to the Author

Reviewer #1: This study investigates the intraspecific variation of specific leaf area (SLA), attributed to edaphic and topographic conditions with a focus on active-layer depth (not “permafrost depth”, as shown in the title). They showed relationships between SLA and active-layer depth through the edaphic environment and pointed out the influence of future permafrost thaw on tree productivity (NPP). The introduction of permafrost characteristics to plant trait modeling is the novelty of this study, and their contribution to improving ecosystem models is expected.

Aim of study and results are clear, and description and illustration of results are mostly clear and well structured. In the conclusion, however, the authors should give remark corresponding to their hypotheses given in introduction. Although critical modification is not necessary through the manuscript, in the method section there is some insufficient description, which should be complemented (please see specific comments).

Based on these evaluations, my recommendation is to ask for a minor revision.

Specific comments

The title of the manuscript is somewhat hard to catch. What is the agent of “control”? “permafrost depth” might be replaced to “active-layer depth”.

L.18: What is “forest composition”?

L.55–59: In this paragraph, the importance of intraspecific SLA variation is introduced. Then, the next paragraph (L.60–65) is back to environmental effect on SLA (focus on permafrost), and the next (L.66–) is devoted to ecosystem modeling and its requirement of intraspecific SLA variation. I think the contents of these paragraphs (L.55–59 and L.60–65) might reverse its order.

L.91: Is Picea glauca white spruce?

L.95: A brief explanation of “cyclic sampling design” helps the reader to understand the experimental design.

L.97: It should be described here explicitly that this transect site locates facing east (L.385). I confused with the description in L.90–94, in which vegetation information is given as north- and south-facing slope. Distribution gradient from white spruce to black spruce follows west to east transect (is it wright? Inferred from L. 103 and 161). As for deciduous trees, only alders are found in this site? (Later, I wonder why the only alder is measured among deciduous trees). And do they stand on upslope or mixing with spruce? Please add information on vegetation.

L.101, 104: Are these values sum of spruce and alder? What does the range (±) indicate? Do you measure at several points along transects?

L.111: “with up to 10 subsamples per species and location”

I cannot catch this phrase. What is subsamples? Please consider rewrite this to clarify.

L.129: sampling location -> soil sampling location

L.135: Is organic layer thickness is included in depth of the moss layer (as caption of Fig. 7)?

L.169: “contiuous data” -> high-resolution spatial data?

L.188: Is this first result (Table1) based on tree (individual) average (n=tree number), not value of leaf (n=total leaf number)? Please explicitly mention.

L.211: Please add explanation of blue line and shade area. Also for the following figures.

L.264: rho -> ρ (Greek letter)

L.299: “a two-fold increase in black spruce SLA corresponds to a five-fold increase in NPP”

I cannot catch this description (ground of “two-fold” and “five-fold”) from Figure 2a. Please add explanation.

L.307: soil’s active layer -> active layer

L.346–349: This sentence seems complicated (duplicate while-clauses). Please consider revise it.

L.353: “bivariate regression” Is this same to “univariate regression” found in Line 175 and 363?

L.358–370: I wonder if this paragraph, in which that not conducted in this study (i.e., SEM analysis) is presented, provides productive discussion here. (I don’t intend to delete it).

L.375: What is “all spatial and temporal scales”? Probably, indicated ecological scales (e.g., Messier et al., 2010)? Please supplement them in a little more detail.

L.386: permafrost not encountered -> not encountered at 1.5m depth

L.394: What does threshold of ALD (dividing shallow and deep active layer) indicate physically? Will you add discussion of possible mechanism how saturation condition is constantly present with ALD shallower than 140cm (not gradually decreasing soil water as deeper active layer)?

L.397–: Again, organic layer is included in the moss layer? Adding to the moss layer, organic layer has also an important role as surface insulation (e.g., Fisher et al., 2016, Johnson et al., 2013, doi:10.1088/1748-9326/8/3/035028). Furthermore, the insulation effectiveness of the moss/organic layer will change due to soil wetness (e.g., O’Donnell et al., 2009, doi: 10.1097/ss.0b013e3181c4a7f8).

S1: What is “REML”?

S1 Figure: How about indicating all kind of sampling (measurement) point including ALD and tree rings?

S2 Figure: Please add explanation of blue line and shade area.

Reviewer #2: Specific leaf area (SLA) influences canopy expansion and growth through its effect on total leaf area per plant affecting light interception and light use efficiency. Investigating the variation of SLA in permafrost region is important to address how soil active layer above permafrost influenced SLA. However, there are some major flaws that prevents the manuscript published in current form.

1. The short title of the manuscript is “Specific leaf area variation with permafrost depth in a boreal forest”. I thought the highlight of the manuscript is to show how permafrost depth influences SLA variation. However, the manuscript showed the how soil conditions (such as soil C:N, ALD) influenced SLA variations. I think the title did not properly represent the content of the manuscript.

2. In Fig.9, the authors used a conceptual model to show the relationship between different soil conditions and SLA changes. Why did the authors give a form Structure Equation Model (SEM) to show this diagram as the authors had the data? The SEM can prove whether the conceptual model is true or not.

3. The relationships showed in Fig3a, Fig3b, Fig.5 and Fig.7 are questionable. The fitted models are highly influenced by the extreme values.

6. PLOS authors have the option to publish the peer review history of their article (what does this mean?). If published, this will include your full peer review and any attached files.

Reviewer #1: No

Reviewer #2: No

---

## [Author Response · Author response to Decision Letter 0]

6 Nov 2020

Dear Editor and Reviewers,

We appreciate your constructive feedback and comments to improve our manuscript. We have addressed each point below (in italics).

Please let us know if further revisions are requested.

Sincerely,

Carolyn Anderson (cganderson@umass.edu), on behalf of my co-authors

Editor Comments

Response: We have addressed this in our Methods section.

Reviewer #1 Comments

Reviewer #1: This study investigates the intraspecific variation of specific leaf area (SLA), attributed to edaphic and topographic conditions with a focus on active-layer depth (not “permafrost depth”, as shown in the title). They showed relationships between SLA and active-layer depth through the edaphic environment and pointed out the influence of future permafrost thaw on tree productivity (NPP). The introduction of permafrost characteristics to plant trait modeling is the novelty of this study, and their contribution to improving ecosystem models is expected.

Aim of study and results are clear, and description and illustration of results are mostly clear and well structured. In the conclusion, however, the authors should give remark corresponding to their hypotheses given in introduction. Although critical modification is not necessary through the manuscript, in the method section there is some insufficient description, which should be complemented (please see specific comments).

Based on these evaluations, my recommendation is to ask for a minor revision.

Specific comments

The title of the manuscript is somewhat hard to catch. What is the agent of “control”? “permafrost depth” might be replaced to “active-layer depth”.

Response: The title was changed to better reflect the findings.

L.18: What is “forest composition”?

Response: “Forest composition” was changed to “dominant tree species”

L.55–59: In this paragraph, the importance of intraspecific SLA variation is introduced. Then, the next paragraph (L.60–65) is back to environmental effect on SLA (focus on permafrost), and the next (L.66–) is devoted to ecosystem modeling and its requirement of intraspecific SLA variation. I think the contents of these paragraphs 

(L.55–59 and L.60–65) might reverse its order.

Response: We switched the order of paragraphs to improve the flow of information about the causes of SLA variation and implications on ecosystem modeling.

L.91: Is Picea glauca white spruce?

Response: White spruce was added in the sentence.

L.95: A brief explanation of “cyclic sampling design” helps the reader to understand the experimental design.

Response: A brief explanation of cyclic sampling design was added to clarify this concept to the reader.

L.97: It should be described here explicitly that this transect site locates facing east (L.385). I confused with the description in L.90–94, in which vegetation information is given as north- and south-facing slope. Distribution gradient from white spruce to black spruce follows west to east transect (is it wright? Inferred from L. 103 and 161). As for deciduous trees, only alders are found in this site? (Later, I wonder why the only alder is measured among deciduous trees). And do they stand on upslope or mixing with spruce? Please add information on vegetation.

Response: The field site description was edited to clarify vegetation transitions over space, including where species occur or co-occur. The reasons for sampling spruce and alder in this study were added.

L.101, 104: Are these values sum of spruce and alder? What does the range (±) indicate? Do you measure at several points along transects?

Response: The text was edited to clarify that stand density and basal area were calculated across the transects and inclusive of several species. The text was also edited to make clear that standard deviation is presented.

L.111: “with up to 10 subsamples per species and location”

I cannot catch this phrase. What is subsamples? Please consider rewrite this to clarify.

Response: This sentence was edited to clarify that subsamples are leaf replicates from an individual tree.

L.129: sampling location -> soil sampling location

Response: This sentence was edited to clarify soil sample location.

L.135: Is organic layer thickness is included in depth of the moss layer (as caption of Fig. 7)?

Response: Only the moss depth is included, and thus we updated the caption of Fig. 7.

L.169: “contiuous data” -> high-resolution spatial data?

Response: The word “continuous” was replaced with “high-resolution spatial data” to clarify the spatial aspect of these data.

L.188: Is this first result (Table1) based on tree (individual) average (n=tree number), not value of leaf (n=total leaf number)? Please explicitly mention.

Response: This sentence was edited to clarify that SLA values were averaged by tree individual for each species, with n=number of trees.

L.211: Please add explanation of blue line and shade area. Also for the following figures.

Response: Text was added to each figure (as needed) to explain the blue line and shaded area.

L.264: rho -> ρ (Greek letter)

Response: Here are elsewhere in the manuscript the word “rho” was replaced by the corresponding Greek letter.

L.299: “a two-fold increase in black spruce SLA corresponds to a five-fold increase in NPP”

I cannot catch this description (ground of “two-fold” and “five-fold”) from Figure 2a. Please add explanation.

Response: This sentence was edited to increase clarity about shifts in SLA and NPP.

L.307: soil’s active layer -> active layer

Response: The word “soil’s” was deleted from this sentence.

L.346–349: This sentence seems complicated (duplicate while-clauses). Please consider revise it.

Response: This sentence was edited to remove the duplicate while-clauses.

L.353: “bivariate regression” Is this same to “univariate regression” found in Line 175 and 363?

Response: Bivariate regressions (regressing two data types against each other) was performed – the two sentences were edited to reflect this.

L.358–370: I wonder if this paragraph, in which that not conducted in this study (i.e., SEM analysis) is presented, provides productive discussion here. (I don’t intend to delete it).

Response: We decided to keep this paragraph, especially since Figure 9 seems like it would lend itself to an SEM analysis and we would like to provide the reader an explanation of why we did not pursue SEM.

L.375: What is “all spatial and temporal scales”? Probably, indicated ecological scales (e.g., Messier et al., 2010)? Please supplement them in a little more detail.

Response: Information was added to this sentence to clarify what is meant by ecological scales of variation.

L.386: permafrost not encountered -> not encountered at 1.5m depth

Response: The active layer probe length of 150 cm was added for clarity.

L.394: What does threshold of ALD (dividing shallow and deep active layer) indicate physically? Will you add discussion of possible mechanism how saturation condition is constantly present with ALD shallower than 140cm (not gradually decreasing soil water as deeper active layer)?

Response: The text was updated to discuss possible mechanisms of relationships between shifts in ALD and temperature and moisture regimes.

L.397–: Again, organic layer is included in the moss layer? Adding to the moss layer, organic layer has also an important role as surface insulation (e.g., Fisher et al., 2016, Johnson et al., 2013, doi:10.1088/1748-9326/8/3/035028). Furthermore, the insulation effectiveness of the moss/organic layer will change due to soil wetness (e.g., O’Donnell et al., 2009, doi: 10.1097/ss.0b013e3181c4a7f8).

Response: The text was edited to clarify that both moss and organic layers can affect thermal regulation of soils and thus active layer depth, but that our data show only moss depth. We appreciate your citation suggestions and have added them to this paragraph where appropriate.

S1: What is “REML”?

Response: The REML abbreviation was defined in the supporting information.

S1 Figure: How about indicating all kind of sampling (measurement) point including ALD and tree rings?

Response: In this figure, we especially wanted to highlight the cyclic design used for soil sampling. The SLA data provides the reader context of how SLA sampling overlaid this design. To account for your comment, we added text to this figure to remind the reader the frequency of the ALD and tree core data.

S2 Figure: Please add explanation of blue line and shade area.

Response: The figure caption was clarified to explain what the blue line and shade area represent.

Reviewer #2 Comments

Reviewer #2: Specific leaf area (SLA) influences canopy expansion and growth through its effect on total leaf area per plant affecting light interception and light use efficiency. Investigating the variation of SLA in permafrost region is important to address how soil active layer above permafrost influenced SLA. However, there are some major flaws that prevents the manuscript published in current form.

1. The short title of the manuscript is “Specific leaf area variation with permafrost depth in a boreal forest”. I thought the highlight of the manuscript is to show how permafrost depth influences SLA variation. However, the manuscript showed the how soil conditions (such as soil C:N, ALD) influenced SLA variations. I think the title did not properly represent the content of the manuscript.

Response: We have changed the title to better reflect the focus of this paper.

2. In Fig.9, the authors used a conceptual model to show the relationship between different soil conditions and SLA changes. Why did the authors give a form Structure Equation Model (SEM) to show this diagram as the authors had the data? The SEM can prove whether the conceptual model is true or not.

Response: We agree that an SEM would be a valuable tool to confirm our conceptual model. We have already provided a comment about why we were not able to do an SEM (limited amount of data to run such a model), and provide suggestions for further studies to increase the sample size to allow for SEM analysis.

3. The relationships showed in Fig3a, Fig3b, Fig.5 and Fig.7 are questionable. The fitted models are highly influenced by the extreme values.

Response: To address this concern, we performed Theil-Sen regression (using R package mblm) as a robust estimator against outliers. Figures and statistics in the Results section were updated to reflect this approach.

---

## [Editor Report · Decision Letter 1]

10 Nov 2020

Active layer depth and soil properties impact specific leaf area variation and ecosystem productivity in a boreal forest

PONE-D-20-10890R1

Dear Dr. Anderson,

We’re pleased to inform you that your manuscript has been judged scientifically suitable for publication and will be formally accepted for publication once it meets all outstanding technical requirements.

Kind regards,

Dafeng Hui, Ph.D.

Academic Editor

PLOS ONE

Additional Editor Comments (optional):

The authors have adequately addressed the reviewers' concerns.
---

## [Editor Report · Acceptance letter]

21 Dec 2020

PONE-D-20-10890R1 

Active layer depth and soil properties impact specific leaf area variation and ecosystem productivity in a boreal forest 

Dear Dr. Anderson:

I'm pleased to inform you that your manuscript has been deemed suitable for publication in PLOS ONE. Congratulations! Your manuscript is now with our production department. 

Kind regards, 

on behalf of

Dr. Dafeng Hui 

Academic Editor

PLOS ONE